# Effects of Caffeine Dose and Administration Method on Time-Trial Performance: A Systematic Review and Network Meta-Analysis

**DOI:** 10.3390/nu17233792

**Published:** 2025-12-03

**Authors:** Ruiguo Xue, Jin Huang, Bin Chen, Li Ding, Li Guo, Yinhang Cao, Olivier Girard

**Affiliations:** 1School of Athletic Performance, Shanghai University of Sport, Shanghai 200438, China; 2321811027@sus.edu.cn (R.X.); 2421852025@sus.edu.cn (J.H.); 2321111021@sus.edu.cn (L.D.); 2Department of Public Physical Education, Fujian Agriculture and Forestry University, Fuzhou 350002, China; chenbin@fafu.edu.cn; 3School of Exercise and Health, Shanghai University of Sport, Shanghai 200438, China; guoli@sus.edu.cn; 4School of Human Sciences (Exercise and Sport Science), The University of Western Australia, Perth 6009, Australia; oliv.girard@gmail.com

**Keywords:** caffeine supplementation, administration method, dosage, endurance performance, ergogenic aids, self-paced exercise

## Abstract

**Background/Objectives:** Caffeine is a well-established ergogenic aid for endurance performance. However, the optimal intake strategy, specifically the administration method and dosage, remains uncertain. This systematic review and network meta-analysis compared the effectiveness of different caffeine administration methods and dosages on time-trial performance. **Methods:** A systematic review and network meta-analysis were conducted following PRISMA guidelines. A systematic search of PubMed, Embase, Web of Science, Scopus, and SPORTDiscus was conducted up to July 2025. Eligible studies were independently screened and quality-assessed by two reviewers. Pairwise and network meta-analyses were conducted to examine the effects of caffeine administration methods (e.g., capsules/tablets, gum, mouth rinse) and dosages (low: ≤3 mg/kg; moderate: 4–6 mg/kg) on time-trial performance. **Results:** Forty-eight studies with 612 participants were included. Low-dose capsules most effectively reduced completion time (standardized mean differences [SMD] = −0.34; 95% confidence interval [CI] −0.62, −0.06), followed by moderate-dose capsules (SMD = −0.31; 95% CI: −0.45, −0.17) and moderate-dose gum (SMD = −0.30; 95% CI: −0.57, −0.02). Low-dose capsules also had the highest probability of improving mean power output (SMD = 0.38; 95% CI: 0.09, 0.67), with moderate-dose capsules ranking second (SMD = 0.30; 95% CI: 0.12, 0.48). **Conclusions:** This systematic review and network meta-analysis identified low-dose caffeine capsules (≈3 mg/kg) as the most effective strategy for improving time-trial performance, with moderate-dose capsules and gum serving as viable alternatives. While these findings provide robust, actionable evidence for practitioners, meaningful inter-individual variability persists. Accordingly, future studies should integrate deeper mechanistic profiling (e.g., genetics and body composition) to advance personalized, evidence-based caffeine supplementation for athletes.

## 1. Introduction

Caffeine (1,3,7-trimethylxanthine) is one of the most widely used ergogenic aids, enhancing performance in endurance [1,2], resistance [3,4], and sprint exercises [5,6]. Among these, endurance sports report the highest prevalence of caffeine use [7]. Research investigating the effects of caffeine ingestion on endurance performance has primarily relied on two protocols: time to exhaustion and time trials. Of these, the time-trial tests are considered more ecologically valid as they better replicate real-world competition and produce findings that are more directly applicable to coaches and athletes [8].

Our previous meta-analysis demonstrated that a moderate caffeine dose (4–6 mg/kg) significantly improves time-trial performance compared to a low dose (1–3 mg/kg) [9]. However, several limitations restrict the generalizability of these findings: (1) only cyclists were included, which limits applicability to other endurance sports (e.g., running, swimming, rowing); (2) only three studies investigated low-dose caffeine (≤3 mg/kg) [10,11,12], reducing confidence in identifying the optimal dosage; and (3) emerging caffeine administration methods (e.g., chewing gum, mouth rinse) were not considered. Recent studies suggest that alternative delivery methods may offer advantages over conventional traditional capsules or tablets, such as faster absorption rates and fewer side effects [13,14,15]. These benefits may in turn enhance ergogenic potential. Therefore, incorporating novel administration methods into analysis is essential for a more comprehensive evaluation of caffeine strategies to enhance endurance performance, particularly time trials.

Several studies have compared different caffeine administration methods on endurance performance, yet findings remain inconsistent [14,16,17]. For instance, direct evidence indicates that 3 mg/kg caffeine tablets reduced 5 km running time by ~2%, whereas gum at this dose shows no effect [14]. Conversely, indirect comparisons report ergogenic potential at higher gum doses (5 mg/kg), which may exceed capsules, with larger performance gains for 5 km completion time (3.6% vs. 1.0%) [16,18]. This inconsistency likely reflects administration-specific dose sensitivities rather than superiority of one format alone. Furthermore, investigations into caffeine mouth rinsing have shown benefits such as increased distance covered in 30 min time trials [19,20]. However, direct head-to-head comparisons among mouth rinse, capsules, and caffeinated gum remain limited. Therefore, a network meta-analysis is warranted to integrate both direct and indirect evidence, allowing a more comprehensive evaluation of caffeine administration methods and dosages to identify the optimal supplementation strategy for endurance performance.

This study employed a network meta-analysis to systematically compare the effects of different caffeine administration methods (capsules/tablets, chewing gum, and mouth rinse) and dosages (low: ≤3 mg/kg; moderate: 4–6 mg/kg) on time-trial performance. The primary aim was to identify the most effective administration method and dosage. These findings offer practical guidance for athletes and coaches in selecting optimal caffeine supplementation strategies.

## 2. Methods

### 2.1. Protocol and Registration

This meta-analysis was conducted following the Preferred Reporting Items for Systematic Reviews and Meta-Analyses extension statement for Network Meta-Analyses (PRISMA-NMA) (Appendix A) [21] and the Cochrane Handbook for the Systematic Review of Interventions [22]. The protocol was prospectively registered in the International Prospective Register of Systematic Reviews (registration number: CRD42024579996). As this research involved the synthesis of previously published data, ethics review board approval and participant informed consent were not required.

### 2.2. Search Strategy and Study Selection

A systematic search was conducted in PubMed, Embase, Web of Science, Scopus, and SPORTDiscus to identify research articles evaluating the effects of different caffeine administration methods and dosages on time-trial performance. All English-language articles published worldwide from database inception to July 2025 were considered. To ensure completeness, the reference lists of relevant reviews and meta-analyses were also manually screened. Search terms (operators) were combined with Boolean conjunctions (OR/AND) and applied across three levels (the full search strategy is presented in Appendix A). Two reviewers (HJ and XR) independently screened titles and abstracts, followed by full-text assessment against the inclusion criteria. Any disagreements were resolved through discussion with a third researcher (CY). All records were managed using Endnote X9 (Thompson ISI Research Soft, Philadelphia, PA, USA), which facilitated screening, citation management, and removal of duplicates.

### 2.3. Eligibility Criteria

Studies were included if they met the following conditions: (1) Participants: healthy adults (>18 years) without cognitive, neurological, orthopaedic, and/or cardiac conditions that could affect physical testing and training; (2) Intervention: isolated caffeine administration via capsules/tablets, chewing gum, or mouth rinse, at low (≤3 mg/kg [23]) or moderate (4–6 mg/kg [9]) doses; (3) Comparator: a placebo group was included; (4) Outcomes: time-trial performance assessed using completion time and/or mean power output; (5) Study design: single- or double-blind, placebo-controlled crossover trials. Studies were excluded if they: (1) grouped participants by genotype based on caffeine sensitivity [9,24]; (2) utilized energy drinks during exercise; (3) used caffeine doses < 2 mg/kg, as such amounts show insufficient ergogenic effects on time-trial performance [25]; (4) were conducted in extreme environments (e.g., high altitude, high or low temperatures) if the caffeine and placebo trials did not occur under the same extreme conditions [24]; (5) were published only as non-English full texts.

### 2.4. Data Extraction

Two reviewers (HJ and XR) independently extracted data from all studies that met the inclusion criteria. Any disagreements were resolved by consulting a third reviewer (CY). Extracted data included: (1) study details (i.e., first author, year of publication); (2) participant characteristics (i.e., age, sex, sample size, maximal oxygen uptake, habitual caffeine intake); (3) intervention characteristics (i.e., caffeine dose, administration method, ingestion timing); and (4) exercise protocol and main findings. If data were unavailable, corresponding authors were contacted up to three times in three weeks. For studies reporting only means and standard deviations (SD) in figures, WebPlotDigitizer Version 4 (Free Software Foundation, Boston, MA, USA) was used to extract values [26]. If SD values were not provided, they were derived from standard errors, 95% confidence intervals, *p* values, or *t* statistics [22].

### 2.5. Risk of Bias Assessment

The risk of bias (RoB) for all included studies was independently evaluated by two investigators (HJ and XR) using the Cochrane Collaboration’s risk-of-bias tool (RoB version 2.0, based in London, UK) [27]. Any disagreements were resolved through consultation with a third experienced reviewer (CY). The tool evaluated bias across the following domains: randomization process; deviations from intended interventions; missing outcome data; outcome measurement; selection of the reported result; and overall bias risk.

### 2.6. Statistical Analysis

For each outcome, a pairwise meta-analysis of available direct comparisons was performed using a random-effects model. Effect sizes were pooled as standardized mean differences (SMD) with corresponding 95% confidence interval (CI). SMDs were calculated as the difference between group means divided by the pooled standard deviation, and interpreted as follows: trivial (<0.20), small (0.20–0.49), moderate (0.50–0.79), and large (≥0.80) [28]. Heterogeneity was assessed using the *I*^2^ test, and categorized as none (0%), low (25%), moderate (50%), or high (75%) [29].

The assumption of transitivity was first assessed by comparing intervention characteristics and the participants’ baseline data to ensure valid pooling across studies [30]. Subsequently, a network meta-analysis was performed using a frequentist graph-theoretical model implemented in the R package net meta (version 4.1.1; The R Foundation for Statistical Computing, Vienna, Austria). Treatment effects were estimated via weighted least-squares regression with the Moore–Penrose pseudoinverse method [31], and between-study variance was modelled using the DerSimonian–Laird random-effects estimator [32]. Network evidence diagrams were constructed, where nodes represented interventions, line thickness reflected the number of studies, and node size was proportional to sample size. Forest plots and league tables of the relative treatment effects were used to visualise comparisons of network estimations. Interventions were ranked according to p-scores, reflecting the certainty that one treatment was superior to another [33]. Decomposed Q-statistics (within and between designs) were used to interpret potential heterogeneity and inconsistency. Heterogeneity and inconsistency were quantified using *I*^2^ [34]. Local inconsistency of direct and indirect results was assessed with the side-splitting method [35]. Publication bias was assessed using comparison-adjusted funnel plots and Egger’s test [36]. Random-effects meta-regression analyses were conducted based on age, maximal oxygen uptake, ingestion timing, habitual caffeine intake, exercise mode, duration, and distance. Network meta-regression was performed with the “Gemtc” package in R software.

## 3. Results

### 3.1. Study Selection and Characteristics

A total of 3502 articles were retrieved from the electronic databases. After removal of duplicates and title/abstract screening, 102 studies underwent full-text review, of which 61 were excluded. Finally, 48 studies were included in the network meta-analysis (Figure 1). Together, these studies included 612 participants, of whom 546 were male (89%). Participant mean ages ranged from 18 to 48 years, and mean $\dot{V}$O_2max_ values ranged from 31.9 to 72 mL/min/kg. Caffeine doses varied between 2 and 6 mg/kg of body weight. The most common intervention was caffeine capsules (n = 35), followed by caffeinated gum (n = 8) and caffeine mouth rinse (n = 5). A detailed summary of each study is presented in Appendix A.

### 3.2. Risk of Bias

Of the 48 studies included, none were judged to be at high risk of bias. A total of 39 were rated as having some concerns, while 9 were assessed as low risk of bias. Full details of the risk of bias assessment are shown in Appendix A.

### 3.3. Network Meta-Analysis

The network evidence plot for completion time is presented in Figure 2a. In the pairwise meta-analysis of completion time, low-dose capsules (SMD = −0.34, 95% CI = −0.62, −0.06; *I*^2^ = 0%), moderate-dose capsules (SMD = −0.31, 95% CI = −0.44, −0.17; *I*^2^ = 0%), and moderate-dose gum (SMD = −0.28, 95% CI = −0.57, −0.01; *I*^2^ = 0%) showed each produced significant reductions in completion time compared to the placebo control group, with consistently low heterogeneity (<25%) (Table 1 and Appendix A). In a network meta-analysis, low-dose capsules (p-score 68%; SMD = −0.34, 95% CI = −0.62, −0.06), moderate-dose capsules (p-score 64%; SMD = −0.31, 95% CI = −0.45, −0.17), and moderate-dose gum (p-score 61%; SMD = −0.30, 95% CI = −0.57, −0.02) all significantly reduced completion time relative to controls (Figure 2b and Table 1). These results indicate strong consistency between the pairwise and network meta-analyses, reinforcing the robustness of the observed effects.

The network evidence plot for mean power output is shown in Figure 2a. In the pairwise meta-analysis of mean power output, both low-dose capsules (SMD = 0.38, 95% CI = 0.09, 0.67; *I*^2^ = 0%) and moderate-dose capsules (SMD = 0.30, 95% CI = 0.12, 0.48; *I*^2^ = 0%) significantly increased mean power output compared to the placebo control group, with low heterogeneity (<25%) across studies (Table 1 and Appendix A). In the network meta-analysis, low-dose capsules (p-score 73%; SMD = 0.38, 95% CI = 0.09, 0.67) and moderate-dose capsules (p-score 62%; SMD = 0.30, 95% CI = 0.12, 0.48) were significantly more effective than control in increasing mean power output (Figure 2b and Table 1). Therefore, the results of the pairwise and network meta-analyses were generally consistent.

We analyzed completion time and mean power output, with the networks showing low heterogeneity and no significant inconsistency (see Appendix A). Examination of baseline characteristics (sample size, mean age, $\dot{V}$O_2max_, and caffeine withdrawal) supported the validity of the transitivity assumption (see Appendix A). The funnel plots revealed poor symmetry around the zero line for completion time and mean power output outcomes, indicating possible publication bias or small-study effects. This interpretation is supported by the significant Egger’s test (*p* < 0.01) (see Appendix A). After conducting a trim-and-fill analysis to assess the impact of publication bias, the overall time-trial performance effects remained significant. This indicates that the results were not significantly influenced by publication bias and can therefore be considered reliable (see Appendix A). Network meta-regression indicated no significant associations between effect sizes (for both completion time and mean power output) and potential moderators including age, $\dot{V}$O_2max_, ingestion timing, habitual caffeine intake, exercise mode, duration, or distance (see Appendix A).

## 4. Discussion

This study presents the first network meta-analysis to systematically compare the effects of different caffeine administration methods (capsules/tablets, chewing gum, and mouth rinse) and dosages (low: ≤3 mg/kg; moderate: 4–6 mg/kg) on time-trial performance, assessed through completion time and mean power output. The findings indicate that low-dose capsules (≈3 mg/kg) were the most effective intervention for reducing completion time, followed by moderate-dose capsules and moderate-dose gum. Likewise, low-dose capsules (primarily 3 mg/kg) had the highest probability of improving mean power output, with moderate-dose capsules ranking second.

Our network analysis identified caffeine capsules as producing the greatest reduction in time-trial completion time among all delivery forms (Figure 2 and Table 1). This finding aligns with previous direct-comparison research showing that 3–4.5 mg/kg caffeine capsules significantly reduced exercise time, whereas caffeinated gum at the same dose did not [14]. One explanation is that residual caffeine often remains in discarded gum, resulting in lower serum caffeine concentrations compared to capsules and thus weaker ergogenic effects, particularly during prolonged exercise [37,38]. Furthermore, the ergogenic effect of capsules may be amplified by factors such as familiarity or prior beliefs, similar to a placebo-like response. Likewise, our findings suggest that caffeine mouth rinse has limited impact on time-trial performance (Figure 2 and Table 1), consistent with recent studies reporting negligible or non-significant effects on completion time [39,40,41,42]. This may be due to the short contact time (5–10 s) of the rinse with the oral mucosa, which allows only trace amounts of caffeine to enter systemic circulation, likely insufficient to elicit meaningful ergogenic effects [23,40]. In addition, studies report limited benefits of caffeine mouth rinse for endurance and high-intensity exercise performance [43]. This may reflect low caffeine absorption during rinsing, as meaningful ergogenic effects typically require higher absorbed dose, more consistently achieved by ingesting the solution rather than rinsing alone [44,45]. Moreover, liquid supplements generally produce stronger placebo responses than capsules, suggesting that the placebo effect may account for a greater proportion of the observed performance improvement [46]. Consequently, the taste-mediated placebo component of mouth-rinse placebos may be a key factor when evaluating true efficacy, whereas placebo capsules (i.e., providing minimal gustatory stimulation) may elicit a smaller placebo response [46]. Overall, caffeine capsules appear to be the most effective administration form for improving time-trial performance, outperforming other caffeine consumption forms.

Our findings show that moderate-dose (4–6 mg/kg) and low-dose caffeine capsules (primarily 3 mg/kg) are similarly effective in improving both completion time and mean power output, with no significant differences between the two. However, given the lower incidence of adverse effects (e.g., gastrointestinal discomfort and irregular heart rate) associated with low-dose caffeine [23], this dosage may be preferable when determining the optimal dosing strategy. Dosage selection, however, should also account for individual variability, particularly habitual caffeine intake. Regular consumers of caffeine-containing beverages such as coffee may develop tolerance [47,48], meaning that low doses might not achieve the desired ergogenic effect [49,50]. Altogether, low-dose caffeine capsules (≈3 mg/kg) appear to offer the most practical balance between performance enhancement and tolerability, representing the optimal supplementation strategy for enhancing time-trial performance.

Network meta-regression analysis revealed no significant associations between effect sizes for completion time and mean power output and the independent variables, including age, $\dot{V}$O_2max_, ingestion timing, habitual caffeine intake, exercise mode, duration, and distance. This finding aligns with the meta-analysis by Southward et al. [24], which also found no associations between comparable covariates (e.g., $\dot{V}$O_2max_, exercise duration, and exercise mode) and caffeine efficiency. This suggests that the effects of caffeine interventions on time-trial performance are largely independent of these moderating factors. Nevertheless, the possibility of undetected modest interactions cannot be ruled out, particularly given the relatively small sample sizes, warranting further investigation.

## 5. Practical Implications

Our results demonstrate that low-dose caffeine capsules (≈3 mg/kg) reduced time-trial completion time by 2.2% compared with placebo, outperforming both moderate-dose capsules (1.8%) and moderate-dose gum (1.6%). While this additional 0.4–0.6% improvement may seem modest, it can be decisive in elite self-paced events when outcomes are separated by mere seconds [51]. For instance, in the men’s individual cycling time trial (32.4 km) at the Paris 2024 Olympics, such a difference could have elevated a fourth-place finisher (36:39.95) onto the podium (36:37.79) [52]. Beyond performance, low-dose caffeine regimens are also associated with fewer adverse effects (e.g., palpitations, insomnia, gastrointestinal discomfort), which may benefit both athletic performance and recovery [23]. Accordingly, low-dose caffeine capsules represent the most practical supplementation strategy for endurance athletes. However, given the considerable interindividual variability in caffeine responses, supplementation regimens should be tailored to personal characteristics (e.g., sex [53] and age [54]).

## 6. Strengths and Limitations

A major strength of this study is the use of network meta-analysis, which enables simultaneous comparison of multiple caffeine administration forms and doses within a unified statistical framework. However, several limitations should be acknowledged. First, relatively few studies have investigated the effects of emerging caffeine administration methods (e.g., gels, energy bars, and nasal sprays) on time-trial performance, leaving their efficacy unclear. Second, because some comparisons were based on indirect evidence (whose validity depends on the assumption of transitivity) and several interventions were supported by limited data, the resulting estimates may lack adequate statistical power and robustness. Third, as over 80% of participants in the included studies were male, the findings are primarily male-specific, and their generalizability to female populations requires further investigation. Fourth, the applicability of our findings beyond endurance-sport protocols remains uncertain, as all included studies focused on endurance exercise. It remains unclear whether similar ergogenic benefits extend to other exercise modalities (e.g., resistance exercise or intermittent high-intensity efforts). Fifth, while placebo effects are known to influence endurance performance [55,56], the studies included here were not designed to isolate or quantify expectancy-driven responses. Therefore, we cannot exclude or quantify the magnitude of placebo contributions across caffeine doses and administration methods.

Another important consideration is that our analysis did not account for individual factors (e.g., body composition [57], metabolic differences [58], and genetic variability [59]) that may influence the ergogenic response to caffeine. For instance, individuals with higher body fat percentages may have higher plasma levels of caffeine and its metabolites and process caffeine more slowly than leaner individuals [57,60]. Likewise, age-related changes in metabolic function (i.e., including energy metabolism, lipid metabolism, and glucose metabolism) may affect caffeine absorption, distribution, metabolism, excretion [58]. Genetic factors also contribute to interindividual variability in caffeine metabolism [59]. Carriers of the A/C or C/C alleles tend to metabolize caffeine more slowly, resulting in a longer half-life and potentially prolonged ergogenic effects, whereas faster metabolizers experience quicker clearance and shorter duration of action [24,61,62,63]. Future research should directly test how individual characteristics (e.g., body composition, metabolic differences, and genetic variability) modify time-trial performance response to caffeine supplementation.

## 7. Conclusions

This network meta-analysis indicates that low-dose caffeine capsules (≈3 mg/kg) represent the most effective strategy for improving time-trial performance, followed by moderate-dose capsules and moderate-dose gum. While these findings provide robust evidence to guide athletes and coaches on effective and practical caffeine supplementation strategies, individual variability in response persists. Accordingly, future studies should incorporate more comprehensive analyses (e.g., genetic profiling and body composition) to develop personalized, evidence-based caffeine supplementation approaches for athletes.

## Figures and Tables

**Figure 1 nutrients-17-03792-f001:**
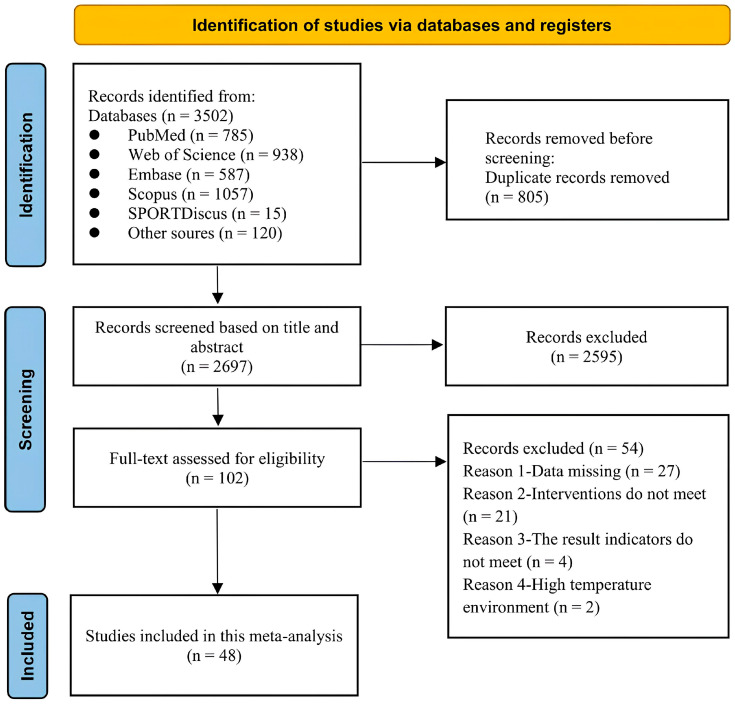
PRISMA flow diagram showing study selection.

**Figure 2 nutrients-17-03792-f002:**
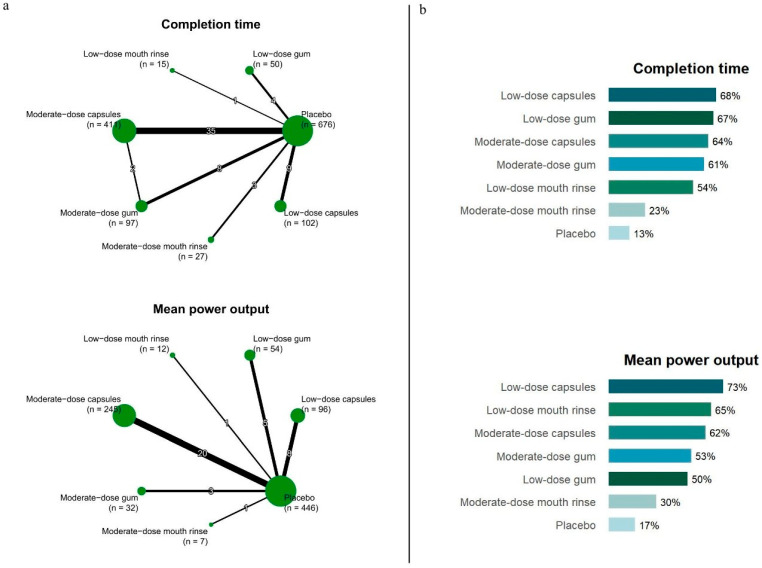
(**a**) Network diagram illustrating the comparative efficacy of different caffeine administration methods and doses on completion time and mean power output. (**b**) p-score rankings illustrating the relative effectiveness of each intervention.

**Table 1 nutrients-17-03792-t001:** League table of the efficacy of different caffeine administration methods and doses on completion time and mean power output.

Completion Time						
Low-dose capsules						**−0.34 (−0.62, −0.06)**
0.00 (−0.48, 0.48)	Low-dose gum					−0.34 (−0.74, 0.05)
−0.03 (−0.34, 0.28)	−0.03 (−0.45, 0.39)	Moderate-dose capsules	−0.00 (−0.52, 0.51)			**−0.31 (−0.44, −0.17)**
−0.05 (−0.43, 0.34)	−0.05 (−0.53, 0.43)	−0.01 (−0.31, 0.28)	Moderate-dose gum			**−0.28 (−0.57, −0.01)**
−0.08 (−0.85, 0.69)	−0.08 (−0.90, 0.74)	−0.05 (−0.78, 0.68)	−0.03 (−0.80, 0.73)	Low-dose mouth rinse		−0.26 (−0.98, 0.46)
−0.34 (−0.94, 0.26)	−0.34 (−1.00, 0.33)	−0.31 (−0.86, 0.25)	−0.29 (−0.89, 0.31)	−0.26 (−1.15, 0.64)	Moderate-dose mouth rinse	−0.00 (−0.54, 0.53)
**−0.34 (−0.62, −0.06)**	−0.34 (−0.74, 0.05)	**−0.31 (−0.45, −0.17)**	**−0.30 (−0.57, −0.02)**	−0.26 (−0.98, 0.46)	−0.00 (−0.54, 0.53)	Placebo
Mean power output						
Low-dose capsules						**0.38 (0.09, 0.67)**
−0.01 (−0.86, 0.85)	Low-dose mouth rinse					0.39 (−0.42, 1.19)
0.08 (−0.26, 0.42)	0.09 (−0.74, 0.91)	Moderate-dose capsules				**0.30 (0.12, 0.48)**
0.14 (−0.43, 0.71)	0.15 (−0.80, 1.09)	0.06 (−0.47, 0.58)	Moderate-dose gum			0.24 (−0.25, 0.73)
0.16 (−0.32, 0.63)	0.16 (−0.73, 1.05)	0.08 (−0.34, 0.49)	0.02 (−0.60, 0.64)	Low-dose gum		0.22 (−0.15, 0.60)
0.43 (−0.65, 1.52)	0.44 (−0.88, 1.76)	0.35 (−0.71, 1.41)	0.29 (−0.86, 1.45)	0.28 (−0.84, 1.39)	Moderate-dose mouth rinse	−0.05 (−1.10, 1.00)
**0.38 (0.09, 0.67)**	0.39 (−0.42, 1.19)	**0.30 (0.12, 0.48)**	0.24 (−0.25, 0.73)	0.22 (−0.15, 0.60)	−0.05 (−1.10, 1.00)	Placebo

Values represent standardized mean differences with 95% credible intervals. Results of the network meta-analysis are presented in the left lower half (blue) and results from the pairwise meta-analysis in the upper right half (green), if available. Grey represents the different interventions. Bold values indicate statistically significant results.

## Data Availability

The original contributions presented in this study are included in the article/Appendix A. Further inquiries can be directed to the corresponding author.

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
