# Peer review of "Effects of Caffeine Dose and Administration Method on Time-Trial Performance: A Systematic Review and Network Meta-Analysis"

_nutrients, 2025, doi:10.3390/nu17233792_

Round 1
Reviewer 1 Report
Comments and Suggestions for Authors
I congratulate the authors on a nice review focusing on low and moderate caffeine dose and TT performance and mean power output.
Generally it is presented well, but I would just like more clarity on the illustration of your results, as I don't quite understand some.
I have just added some points below:
Introduction:
first paragraph - nice start to the introduction and agree with rationale to focus on time trials due to an enhanced validity.
Ln 59-63 - Can you add a comment as to why there may be disagreement within the literature on the highlighted studies?
Methods:
Ln 106 - can you comment as to why participants were excluded if they were grouped based on genotype?
Results:
Ln 203 - can you expand on how publication bias is present within your results
Figure 2a: is the direction and length of the line of interest?
Figure 2b: Don't quite understand the p-score rankings. Can you provide an overview of what these % mean? Placebo is 13-17% effective, is that because the placebo trial improved performance in some studies? Is completion time with low-dose capsules improved 68% of the time?
Table 1: unsure what is meant by the triangle locations. Why are there different number of rows for each intervention. Just unclear.
Discussion:
ln 240: can you explain more here on why the bitter taste from caffeine effects the mouth rinse. Understand that not consuming the drink and fully ingesting will limit the ergogenic effect, but more on your point to taste. In addition, the last sentence on Ln 243 - do you mean 'with the capsule', and is this linked to the flavour?
Ln 252 - state lower dose may reduce adverse side effects, but were these measured or reported?
Author Response
We thank Reviewer 1 for these comments. Please see the attachment.

Reviewer 2 Report
Comments and Suggestions for Authors
The authors sought to examine the effect of caffeine on time trial performance, and the potential role of dose and delivery method using a systematic review and meta-analysis. The authors report that a low dose was equally effective as a moderate dose, and that capsules tended to yield the greatest improvements over other vehicles (e.g. mouth rinse). The study is generally well conducted and presented.
My chief complaint with the paper, at this stage of our knowledge, the authors do not address the placebo effect or the expectancy of caffeine as a means to alter exercise performance. See recent review for example https://pmc.ncbi.nlm.nih.gov/articles/PMC11243088/ as well as others https://pmc.ncbi.nlm.nih.gov/articles/PMC6759201/
Similarly, the authors should try to parse out those that assessed genotypes to explain, at least in part, the heterogeniety of the effects of caffeine on time trial performance.
Without addressing these concerns the paper is not novel as a number of systematic reviews and meta analyses have been written on this topic, so much so that there is an umbrella review.
https://bjsm.bmj.com/content/54/11/681?crsi=6624964158&cicada_org_src=healthwebmagazine.com&cicada_org_mdm=direct
https://link.springer.com/article/10.1007/s40279-018-0939-8
https://journals.humankinetics.com/view/journals/ijsnem/14/6/article-p626.xml
https://link.springer.com/article/10.1007/s00394-016-1331-9
https://journals.humankinetics.com/view/journals/ijspp/13/4/article-p402.xml
Author Response
We thank Reviewer 2 for these comments. Please see the attachment.

Reviewer 3 Report
Comments and Suggestions for Authors
Very well written manuscript overall.
Few comments to address before considering this manuscript for publication:
. Abstract: reference to following PRISMA checklist should be added in the Methods section. In the concluding statement, suggest that authors elaborate on the next steps, and the practical implications of these findings.
Author Response
We thank Reviewer 3 for these comments. Please see the attachment.
